# Budding of a Retrovirus: Some Assemblies Required

**DOI:** 10.3390/v12101188

**Published:** 2020-10-20

**Authors:** Kevin M. Rose, Vanessa M. Hirsch, Fadila Bouamr

**Affiliations:** Laboratory of Molecular Microbiology, National Institute of Allergy and Infectious Diseases, National Institutes of Health, Bethesda, Rockville, MD 20894, USA; kevin.rose@nih.gov (K.M.R.); vhirsch@niaid.nih.gov (V.M.H.)

**Keywords:** ESCRT-I, ALIX, HIV-1 budding, nucleocapsid, ribonucleoprotein, maturation

## Abstract

One of the most important steps in any viral lifecycle is the production of progeny virions. For retroviruses as well as other viruses, this step is a highly organized process that occurs with exquisite spatial and temporal specificity on the cellular plasma membrane. To facilitate this process, retroviruses encode short peptide motifs, or L domains, that hijack host factors to ensure completion of this critical step. One such cellular machinery targeted by viruses is known as the Endosomal Sorting Complex Required for Transport (ESCRTs). Typically responsible for vesicular trafficking within the cell, ESCRTs are co-opted by the retroviral Gag polyprotein to assist in viral particle assembly and release of infectious virions. This review in the Viruses Special Issue “The 11th International Retroviral Nucleocapsid and Assembly Symposium”, details recent findings that shed light on the molecular details of how ESCRTs and the ESCRT adaptor protein ALIX, facilitate retroviral dissemination at sites of viral assembly.

## 1. Introduction

### The Role of Viral L Domains and Host Factors in Virus Assembly

During the final stages of enveloped virus replication, viral proteins are co-assembled with their corresponding viral genome into a heterogeneous ribonucleoprotein complex (RNP). These RNPs are assembled at productive sites of virus release through interactions with many host cell co-factors including the Endosomal Sorting Complex Required for Transport (ESCRT) machinery, with a fraction of these components detected within infectious virions [1,2,3,4]. In order to efficiently transmit the genomic RNP packaged into nascent virions, the Human Immunodeficiency Virus (HIV), as well as other retroviruses, have evolved to encode multiple Late Budding (L) domains within their Gag proteins that recruit specific cellular factors to facilitate release and in turn virus spread. For HIV, the Gag protein encodes two C-terminal L domains with the highly conserved sequences PTAP and LYPXnL, that recruit the ESCRT-I component TSG101 and the ESCRT-I adapter protein, ALIX, respectively [5,6,7,8]. Additionally, some viruses encode just one L domain like the LYPXnL-encoding p9 protein of Equine Infectious Anemia Virus (EIAV) [9]. These two types of L domains and their cognate cellular partners operate in distinct cellular pathways to support vesicular trafficking and therefore independent mechanisms of virus release because abrogation of either individual sequence within HIV Gag still results in infectious particle release, albeit at significantly reduced levels [10,11]. More importantly, L domain deletion displays a compounding negative effect on virus release. In Jurkat and CEM cells, a physiologically relevant T cell system, PTAP (PTAP-) or LYPXnL (YP-) L domain mutant HIV exhibit reductions in virus release of approximately 33 and 50 percent, respectively, while a co-L domain deletion (PTAP-/YP-) showed the greatest release defect of more than 80 percent [12].

In parallel, it was recently shown that cytokinesis, a process that displays many parallels with retroviral budding, depends on cooperativity between both TSG101 and ALIX, potentiating a heterogeneous complex that may be ideal for retrovirus budding, as well as other cellular process, when possible [13]. Both processes of retroviral budding and cytokinesis culminate with a topologically similar membrane abscission process that separates progeny virions from host cell membranes and daughter cells from one another after cell division [13,14,15]. This final event is carried out by the ESCRT-III membrane scission machinery that is recruited by ALIX and ESCRT-I to these scission sites (for review, see [16]). However, this comparison between membrane sealing events also reveals a critical gap in our understanding of ESCRT-III recruitment to these sites. Indeed, the loss of ALIX recruitment to viral budding sites through HIV L domain deletion or to cytokinetic bridges via cellular depletion by RNA interference (RNAi) impairs but does not terminate these events. This mandates the presence of redundant and parallel ESCRT-III recruiting axes perhaps fulfilled by the intermediary ESCRT-II complex [13,17,18] or additional ALIX homologs and orthologs [19,20,21].

Despite their evolutionary conservation between viruses, these short L domain peptide motifs are not the sole determinants of cellular factor recruitment. Recently, it was shown that HIV Gag and other retroviruses are modified with ubiquitin in their p6 domains, and a C-terminal ubiquitin-Gag fusion construct mimicking a true p6 domain modification, can stimulate and rescue defective virus release [22,23]. The N-terminus of TSG101 contains a ubiquitin binding (UEV) domain that is essential for the recognition of Gag via its PTAP sequence, as well as ubiquitinated cargo for multivesicular trafficking, suggesting a functional role for viral Gag ubiquitination [5]. In addition to ubiquitin modification, the Nucleocapsid (NC) region of Gag has been implicated in binding to the Bro1 domain of ALIX, but also in binding to TSG101 [24,25,26,27]. However, the precise molecular details underlying the role of NC in the recruitment and/or the function of these host factors remains unclear, but these interactions are complex and may involve simultaneous binding to RNA [26].

## 2. ESCRT-I Forms Filaments Which Are Essential for the Release of Infectious Virus

In order to elucidate the molecular mechanisms behind the concomitant recruitment of the ESCRT-I complex with viral Gag assembly [28] and subsequent membrane scission, several strides were recently made in understanding the molecular nature of the human and yeast ESCRT-I complexes via X-ray crystallography (Figure 1A,B, respectively). It is now accepted that the functional ESCRT-I complex in humans organizes itself as a hetero-tetramer composed of a single copy of VPS28, TSG101, VPS37, and MVB12. The complex assumes the structure of a globular head that is oriented as a descending stack of alpha helical hairpins with VPS28 on the top and VPS37 on the bottom, with a protruding coiled-coil stalk [29,30,31,32]. Interestingly, we noted that a very similar descending stack of alpha helical hairpins is adopted by the globular head of ALIX known as the Bro1 domain, but no structural or functional significance of this relationship has been identified [33]. Recently, for the first time, the human ESCRT-I complex was solved via crystallography and bore striking similarity to the homologous complex in yeast. In particular, both crystals are composed of helical assemblies of the hetero-tetramer ESCRT-I complexes and a human VPS28-TSG101 subcomplex was shown to form filaments in solution as verified by negative stain electron microscopy [30,32]. Surprisingly, in both yeast and human ESCRT-I filaments, it appeared that contacts between neighboring VPS28 molecules were essential for higher-order assembly (Figure 1A,B).

VPS28 is a 25 kDa protein containing two distinct domains that operate independently to coordinate the events involved in multivesicular body (MVB) trafficking. The N-terminal domain of VPS28 binds to the C-terminal domain of TSG101 while the C-terminal domain of VPS28 binds the ESCRT-II complex, although the role of ESCRT-II in virus release remains debated [17,18,29,34]. The sequence of VPS28 is evolutionarily conserved from yeast to humans. Prior to the structural elucidation of the human ESCRT-I complex, Rose et al. (submitted for review) sought to characterize the role of VPS28 in virus release using structure guided mutagenesis approaches based on human molecular models of the yeast homologues [30]. The yeast-derived human ESCRT-I complexes revealed that a highly conserved motif _54_EKAYIKD_60_ located at the start of the alpha helical hairpin in the N-terminus of human VPS28 was essential for forming helical assemblies, in good agreement with the known yeast complex (Figure 1D). Based on its high sequence conservation across species and its location in the resolved yeast structure, it was likely that this region of the protein played an essential role in ESCRT-I function, and therefore in virus release, by mediating ESCRT-I helical assembly. Indeed, we demonstrated that virus release requires VPS28 with an intact _54_EKAYIKD_60_ sequence because ablation of the motif via alanine scanning abolished the release of genomic RNP-containing virus (manuscript in preparation).

This VPS28 mutant (termed EKYK) was excluded from a WT VPS28-TSG101 complex, suggesting that the _54_EKAYIKD_60_ sequence is essential for proper ESCRT-I assembly and function. More specifically, the VPS28 EKYK mutant failed to capture TSG101 and formed non-productive aggregates with WT VPS28. In contrast, WT VPS28 readily captured TSG101 while forming very little association with the VPS28 EKYK mutant. Ultimately, this suggested that the _54_EKAYIKD_60_ motif in VPS28 was responsible for productive ESCRT-I formation through the ideal multimerization of VPS28 and recruitment of TSG101 to the VPS28 assembly.

At the time of this discovery, Flower and colleagues crystallized the human ESCRT-I complex and remarkably confirmed the _54_EKAYIKD_60_ motif as a critical mediator of ESCRT-I helical assembly. This was done through the identification of a triple mutant (K55D, K59D, and D60A, termed TM) within the _54_EKAYIKD_60_ motif, that prevented the VPS28-TSG101 subcomplex from forming higher order assemblies in solution, which also prevented autophagosome closure [36]. We validated these findings by showing that the VPS28 TM mutant failed to release NC in virus particles. Remarkably, an additional in vitro study recently captured 3–11 copies of TSG101 within budding virions using super-resolution microscopy, in excellent agreement of a single helical turn of ESCRT-I at the base of viral budding necks [35]. These studies shed light on molecular mechanisms of membrane scission closure by fundamentally linking multiple ESCRT-I functions to higher-order assemblies mediated by VPS28 for the first time. ESCRT-I assembly is critical for virus budding and closure of phagophores supporting a universal requirement for multimerization.

## 3. ALIX Bro1-Bro1 Contacts Are Essential for Virus Release

Operating mainly via the second L domain of HIV Gag is ALIX. ALIX is a three-domain protein consisting of an N-terminal Bro1 domain, a central V domain, and a C-terminal proline rich domain (PRD) [33]. The Bro1 and V domains have been extensively characterized as they harbor the ESCRT-III and LYPXnL binding sites, respectively [37,38]. The precise nature of the PRD in ALIX function remains obscure but it has been shown to be a secondary facilitator of ALIX multimerization along with the V domain, to bind TSG101, as well as to recruit additional host factors including endophilin [39,40]. In addition to its role in binding ESCRT-III, the Bro1 domain of ALIX contains two evolutionarily conserved regions termed “surface patches” that are shared by a variety of Bro1 containing proteins. Intriguingly, the region of Bro1 domains called “surface patch 2”, harbors a _317_FIYHDR_321_ motif whose tyrosine and terminal polar residue are completely conserved from yeast to humans. This region within ALIX had been previously identified as a docking site for phosphorylation by Src kinases, but tyrosine phosphorylation is not required for virus release [33,41].

The recent findings pointing toward a helical nature of ESCRT-I prompted us to investigate ALIX in order to determine whether or not the _317_FIYHDR_321_ motif was of structural or functional importance. First, we conducted a structural alignment between the ALIX Bro1 domain and the human hetero-tetramer ESCRT-I model to determine any significant topological similarities based on their curiously similar domain architecture. We found that the C-terminal hairpin of TSG101 strongly aligned with the alpha helical hairpin between helices 8 and 9 of the ALIX Bro1 domain which oriented the _317_FIYHDR_321_ motif about the same axis as the VPS28 multimerization domain. This prompted the revisit of our previously resolved structures of the ALIX Bro1 domain protein and Brox. Surprisingly, we found that the _317_FIYHDR_321_ motif was involved in a multitude of higher-order assemblies, in a manner that was strikingly reminiscent of the VPS28 multimerization domain [38,42]. Crystal structures of Brox alone were found to form trimers via this motif, while they formed dimers about it in the presence of a CHMP5 fragment, and remarkably, lemniscate helices in the presence of a CHMP4B fragment (Figure 1C), the downstream ESCRT component required for membrane scission and virus release (PDB codes: 3R9M, 3UM0, and 3UM3 respectively).

Alanine scanning was performed of the multimerization motif that contained the residues _317_FIY_319_ in ALIX. Virus release was dramatically abrogated for a HIV virus lacking the ESCRT-I L domain (PTAP-) with no detectable release of mature virus. Ablation of virus production was accompanied with dramatic Gag processing defects and the accumulation of Gag maturation intermediate p15NC-p1-p6. The same ALIX mutant was tested for release with EIAV which is solely reliant on ALIX for release. EIAV also displayed budding defects as severe as those observed with HIV-1. These studies show for the first time that the _317_FIY_319_ is linked to the helical assembly of ALIX, as shown via molecular modeling (Figure 1E) and reveal the critical role of such spiral structures in membrane scission and closure.

## 4. Helical Assemblages of the Upstream ESCRTs Are Critical for Virus Maturation by Retention of the RNP

It has been known for over two decades that the L domains encoded by the retroviral Gag polyprotein are critical for the proper packaging of the essential viral processing enzymes required for maturation and infectivity, known as the Gag-Pol polyprotein [43,44,45]. Gag-Pol contains the viral protease which is responsible for cleaving the immature Gag polyprotein into the mature and individually functional components matrix (MA), capsid (CA), and nucleocapsid (NC). When Gag-pol is unincorporated from assembly sites, protease activity is reduced, virus release is impaired and budding virions are marked by a thicker Gag shell that has gone unprocessed [46]. In addition to defects in Gag maturation, budding particles are also often arrested with a large cytoplasmic opening when one or both L domains are deleted, causing significant delays in their maturation up to 10 h [45].

In agreement with this, Rose et al. (submitted for review) observed a substantial increase in processing defects when either the VPS28 or ALIX helical interfaces were disrupted. Remarkably, these same mutants of VPS28 and ALIX also induced severe defects in nuclear envelope repair and delays in cytokinetic abscission. Viral processing and release defects could only be observed by probing cell lysates and virus extracts with antibodies targeting both the N- (CA antibodies) and C-terminal (NC antibodies) regions of Gag. Traditionally, virus processing and release is monitored through the assessment of mature processed CA. However, this approach fails to consider the fate of the RNP assembling on membranes and within the viral particles themselves since processed NC is intimately associated with the RNP. By assessing virus release in such a way, we were able to determine that ESCRT-I and ALIX helical assemblies act to retain the RNP for optimal protease activity within the lumen of budding necks. In doing so, we also demonstrate the requirement for assessing viral maturation products via their NC content in future studies aimed at studying retroviral assembly, for packaging of the RNP ensures that progeny virions will be infectious and capable of establishing a replication-competent reservoir.

Ultimately, we propose a model where ESCRT-I and ALIX form a discontinuous co-polymer at sites of scission to act as a structural scaffold for assembling nascent virions, but also as a template for ESCRT-III polymerization and membrane scission (Figure 2). Such a mechanism is structurally reminiscent of the helical Gamma-Tubulin Small Complex (gamma-TuSC) which acts as a template for microtubule nucleation during cell division [47]. This model was also in good agreement with the recent findings and molecular dynamics simulations by Flower and colleagues where they propose a circular ESCRT-I multimer template for reverse topology membrane scission by the downstream ESCRT-III proteins [32].

## 5. Perspectives

In recent years, several strides have been made in understanding the molecular details underlying Gag assembly and virus budding. In particular, it is the critical advancements in microscopy that have led to these discoveries. The last decade alone has seen astonishing progress with the advent of super-resolution microscopy and exquisite improvements to the capabilities of tomography and electron microscopy. These techniques allowed us to observe molecular details once thought elusive like the spatial and temporal organization of the ESCRTs at sites of membrane scission [4,35,48], or the structures of retroviral Gag lattices as they undergo maturation [49,50,51,52].

Nevertheless, the precise nature of host factors at the budding virion-cytoplasm boundary is yet to be determined, perhaps due to its ephemeral nature. Although small molecule inhibitors have already been developed that are aimed at disrupting this critical interface [53], they interfere with cellular proteins and may be less efficacious than targeting viral proteins directly. In defining the mode of action for ESCRT-I and ALIX as helical polymers in viral scission necks, this mechanism can hopefully guide future studies of related proteins like the Bro1-domain protein HD-PTP that is involved in a mechanistically similar process of neuronal pruning [54]. Additionally, the ESCRT-I-ALIX interaction is essential to cytokinesis and the findings reviewed here may offer insights into the control of cellular abscission, extending a novel understanding of a viral co-option mechanism to that of our own physiology [13].

## Figures and Tables

**Figure 1 viruses-12-01188-f001:**
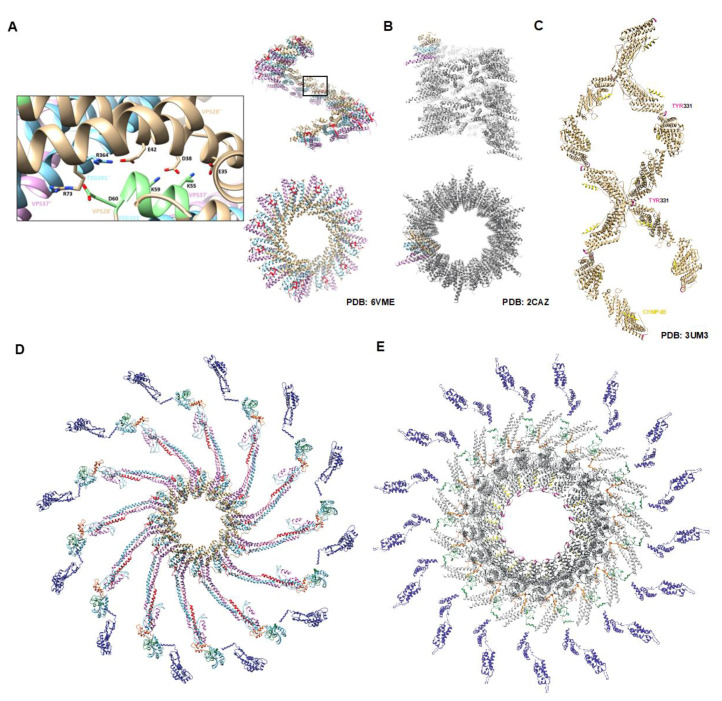
ESCRT-I and ALIX helical assemblies are required for retroviral budding. Crystal structures of human ESCRT-I revealed that the hetero-tetramer core forms filaments mediated principally by contacts within VPS28. (**A**) Filament of human ESCRT-I as determined by X-ray crystallography. VPS28 is colored in beige, TSG101 in blue, VPS37 in pink, and MVB12 in red. Two views of the filament are shown for comparison of the helical and luminal parameters. Inset reveals VPS28 residues essential for filament formation colored in green. (**B**) Filament of the yeast ESCRT-I derived from a crystal (Teo et al., [21]). The individual yeast components are colored the same way as in (**A**) with the exception of MVB12 due to its absence in yeast. Note that the yeast filament adopts a one-start helix and a larger luminal width than that of human ESCRT-I which is a tighter three-start helix. (**C**) Lemniscate helix of Bro-1-domain protein Brox in the presence of a CHMP4B peptide (yellow). Brox is shown in beige with the FIY motif colored in pink at each interface (Mu et al., [35]). (**D**) Molecular modeling derived from the study by Rose et al. (submitted for review) of the yeast-derived human ESCRT-I complex bound to Gag. ESCRT-I components are colored in the same fashion as above and Gag is shown with CA in blue, NC in green, and p6 in orange. The VPS28 multimerization domain is highlighted in light green. (**E**) Luminal view of the molecular model for an ideal full-length ALIX helical assembly bound to Gag. ALIX is shown in grey with its Bro1 multimerization domain in pink. The CHMP4B peptide can be seen between ALIX dimers in gold. Due to differences in length between ALIX and ESCRT-I, the ALIX oligomer adopts a wider luminal width and larger helical repeat (18 copies of ALIX per turn versus 12 for ESCRT-I) in order to accommodate the diameter of assembling retroviral particles.

**Figure 2 viruses-12-01188-f002:**
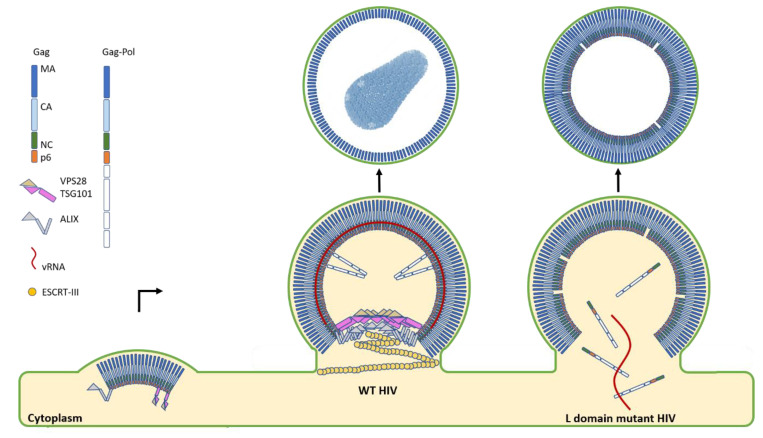
HIV budding diagram depicts the role of the upstream ESCRTs in virion maturation and release. The first step in HIV budding is the clustering of lower-order Gag multimers at the membrane. This occurs with a concomitant recruitment of the ESCRT-I component TSG101 and ALIX to sites of assembly when both L domains are intact. In the case of mutant HIV where one or both L domains are deleted, severe temporal delays in budding are observed. Additionally, particles are found with immature Gag shells, indicative of processing defects due to the leakage of the processing enzymes within the Gag-Pol polyprotein back into the cytoplasm. In stark contrast to this severe budding phenotype, when both L domains are present, ESCRT-I and ALIX are recruited to sites of assembly and remain there during the late stages of maturation. The presence of ESCRT-I and ALIX also coincides with an optimal efficiency of processing by the retroviral protease due to its retention and proper activation within maturing virions. ALIX then recruits the downstream ESCRT-III CHMP4B for the sealing and scission of the budding virion to allow for the rapid and efficient release of progeny virions.

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
