# Peer review of "Budding of a Retrovirus: Some Assemblies Required"

_viruses, 2020, doi:10.3390/v12101188_

Round 1

Reviewer 1 Report

The manuscript is fine, very well written and I recommand publication. But sometimes, it is very specific and on some aspects are suffering of no comparison with other works. For exemple, I recommanded to mention/discuss the recent work from H.deRocquigny's team, on the role of NC/Tsg101/Alix on HIV assembly:

The NC domain of HIV-1 Gag contributes to the interaction of Gag with TSG101.

El Meshri SE, Boutant E, Mouhand A, Thomas A, Larue V, Richert L, Vivet-Boudou V, Mély Y, Tisné C, Muriaux D, de Rocquigny H. Biochim Biophys Acta Gen Subj. 2018 Jun;1862(6):1421-1431. doi: 10.1016/j.bbagen.2018.03.020. Epub 2018 Mar 20. PMID: 29571744

Author Response

Re: Reviewer #1:

Remarks to the Author

We greatly appreciate the feedback and suggestion provided by Reviewer 1.

The manuscript is fine, very well written and I recommend publication. But sometimes, it is very specific and on some aspects are suffering of no comparison with other works. For example, I recommended to mention/discuss the recent work from H. deRocquigny's team, on the role of NC/Tsg101/Alix on HIV assembly: 

The NC domain of HIV-1 Gag contributes to the interaction of Gag with TSG101.

El Meshri SE, Boutant E, Mouhand A, Thomas A, Larue V, Richert L, Vivet-Boudou V, Mély Y, Tisné C, Muriaux D, de Rocquigny H. Biochim Biophys Acta Gen Subj. 2018 Jun;1862(6):1421-1431. doi: 10.1016/j.bbagen.2018.03.020. Epub 2018 Mar 20. PMID: 29571744

We regretfully omitted the reference included in Reviewer 1’s remarks in the initial submission of our manuscript. It is now listed as reference # 23 and was incorporated into the revised text where it is mentioned and compared with several references about the role of NC in binding to TSG101. The reference is cited in the second to last sentence in the second paragraph of section 1 (line 63).

Reviewer 2 Report

Rose, Hirsch, and Bouamr synthesize the current knowledge regarding early acting ESCRT assemblies into a clear and comprehensive review of this aspect of the field. While this reviewer feels that this manuscript is suitable for publication in Viruses MDPI, there are a few missing modern citations and details that should be included to complete this review for a general virology audience. Overall this reviewer felt that there was far less ‘what we don’t know’ than ‘what we do know’. There are still many gaps in our understanding of ESCRT-mediated virus abscission and reviews should excite prospective researchers to this field.

Major Points:

1) Cited in reference 23 (Flower, TG et al.), Hoffman, HK et al (2019) generated and validated CRISPR/Cas9 fluorescent protein knock-in cell lines for the measurement of cellular and viral distributions/levels of Tsg101. This study demonstrates that ~3-11 Tsg101 molecules are incorporated into released HIV-1 particles, consistent with a single helical turn of an ESCRT-I filament demonstrated in Flower, TG et al.

2) The authors propose in Figure 2, middle panel, that the ESCRT-I filament resides at the aperture of the Gag shell. A new model based on the work of Flower, TG et al proposes the ESCRT-I helical filament would place the UEV domains within striking distance of the PTAP motifs lining the aperture of the budding Gag shell. I commend the authors for addressing this spatial context as the link between ESCRT-I and ESCRT-III is often omitted in models (e.g. how does ESCRT-III know where to form filaments at bud sites without templating on upstream ESCRTs). A PTAP deletion makes a perfectly budded particle attached by a thin membrane neck, yet does not abscise even though the topology for ESCRT-III filament formation is perfect. This could be emphasized a bit more if possible.

3) The authors propose that ALIX interaction with the ESCRT-I helical filament recruits the ESCRT-III filament to sites of HIV assembly, yet, loss of ALIX or the LYPXnL motif in HIV Gag has very little effect on ESCRT-mediated particle release if the ESCRT-I binding PTAP motif is left intact. It is only loss of both motifs, followed by overexpression of ALIX can rescue budding defects, suggesting that ALIX plays a minor or alternative role in HIV abscission. The authors must better clarify to a general audience that this discrepancy exists and that there is likely another unidentified player that mediates or plays as surrogate (loss of ALIX) for ESCRT-III recruitment. Could it be other Bro1 domain containing proteins step in with loss of ALIX and this makes the importance of ALIX seem less so in the context of HIV abscission? The authors rightly point out that ESCRT-II’s role in HIV budding is highly debated so this point could be highlighted in that section perhaps.

4) Relatively recent electron microscopy data from Cashikar, AG et al (2014) should be cited showing that small filaments may arrive early in the assembly steps of HIV. Namely Figure 6 from Cashikar, AG et al is likely depicting ESCRT-I filaments or the beginning of ESCRT-III filament assembly at nascent HIV-1 assembly sites. This review should link these findings.

5) The major players for ESCRT-mediated HIV release have been known for quite some time (caveat with point #3), however, how these pieces fit together in supramolecular assemblies to mediate abscission is still unknown yet slowly becoming realized. This reviewer feels that the authors could use a little bandwidth to highlight technologies that are enabling measurements for the organization of the ESCRT machinery at HIV assembly sites or cite suitable references that would educate a virologist audience about these technologies and the challenges of applying them to ESCRT-mediated virus abscission.

Minor Points:

1) The term ‘vestigial’ section 1 (introduction), referring to the catalytically inactive E2-like UEV domain of Tsg101, is not an easily accessible description for a general audience. Consider revision of this sentence to give the audience a better idea of the nature of the UEV domain

2) An ‘et al’ is missing from reference 23

3) There is an underscore at the end of the last sentence/paragraph in section 2.

4) Third sentence, first paragraph, section 3. The ‘N’ should be lowercase.

5) Is the term ‘Infectious’ in the title necessary?

6) I am not sure if there is a page limit for this review, but there could be a bit more ‘what we don’t know’ in the conclusion, this is just a suggestion and not required for revision.

Author Response

We value the critical reading of the manuscript and the constructive suggestions provided by Reviewer 2.

Major Points:

1) Cited in reference 23 (Flower, TG et al.), Hoffman, HK et al (2019) generated and validated CRISPR/Cas9 fluorescent protein knock-in cell lines for the measurement of cellular and viral distributions/levels of Tsg101. This study demonstrates that ~3-11 Tsg101 molecules are incorporated into released HIV-1 particles, consistent with a single helical turn of an ESCRT-I filament demonstrated in Flower, TG et al.

The authors appreciate the suggestion for a reference to Hoffman, HK et al (2019) to be added to the text. These findings are in excellent agreement with the data presented herein. The reference was added as # 36 and was described on line 155.

2) The authors propose in Figure 2, middle panel, that the ESCRT-I filament resides at the aperture of the Gag shell. A new model based on the work of Flower, TG et al proposes the ESCRT-I helical filament would place the UEV domains within striking distance of the PTAP motifs lining the aperture of the budding Gag shell. I commend the authors for addressing this spatial context as the link between ESCRT-I and ESCRT-III is often omitted in models (e.g. how does ESCRT-III know where to form filaments at bud sites without templating on upstream ESCRTs). A PTAP deletion makes a perfectly budded particle attached by a thin membrane neck, yet does not abscise even though the topology for ESCRT-III filament formation is perfect. This could be emphasized a bit more if possible.

A discussion of ESCRT-III topology in the context of retroviral budding and cytokinesis was included in the newly added paragraph 2, section 1, that address this major point by Reviewer 2. This modification begins on line 54 and ends on line 63.

3) The authors propose that ALIX interaction with the ESCRT-I helical filament recruits the ESCRT-III filament to sites of HIV assembly, yet, loss of ALIX or the LYPXnL motif in HIV Gag has very little effect on ESCRT-mediated particle release if the ESCRT-I binding PTAP motif is left intact. It is only loss of both motifs, followed by overexpression of ALIX can rescue budding defects, suggesting that ALIX plays a minor or alternative role in HIV abscission. The authors must better clarify to a general audience that this discrepancy exists and that there is likely another unidentified player that mediates or plays as surrogate (loss of ALIX) for ESCRT-III recruitment. Could it be other Bro1 domain containing proteins step in with loss of ALIX and this makes the importance of ALIX seem less so in the context of HIV abscission? The authors rightly point out that ESCRT-II’s role in HIV budding is highly debated so this point could be highlighted in that section perhaps.

The authors included a more detailed account of reference 12, on line 36, which clearly shows a significant impairment on virus release in Jurkat cells in the absence of ALIX. Additionally, as above, paragraph 2, section 1, was greatly expanded to include a discussion on the scenarios described in this major point by Reviewer 2 (discussion from line 54 to line 63).

4) Relatively recent electron microscopy data from Cashikar, AG et al (2014) should be cited showing that small filaments may arrive early in the assembly steps of HIV. Namely Figure 6 from Cashikar, AG et al is likely depicting ESCRT-I filaments or the beginning of ESCRT-III filament assembly at nascent HIV-1 assembly sites. This review should link these findings.

The reference including in this major point by Reviewer 2 was incorporated into the text in the same modified section as stated above and is listed as reference 15 (line 56).

5) The major players for ESCRT-mediated HIV release have been known for quite some time (caveat with point #3), however, how these pieces fit together in supramolecular assemblies to mediate abscission is still unknown yet slowly becoming realized. This reviewer feels that the authors could use a little bandwidth to highlight technologies that are enabling measurements for the organization of the ESCRT machinery at HIV assembly sites or cite suitable references that would educate a virologist audience about these technologies and the challenges of applying them to ESCRT-mediated virus abscission.

We appreciate the great suggestion brought up by Reviewer 2 in this major point. Section 5, paragraph 1, was greatly modified to include a discussion pertaining to the recent advancements in microscopy that have allowed for the findings reviewed here (line 253 to 258).

Minor Points:

1) The term ‘vestigial’ section 1 (introduction), referring to the catalytically inactive E2-like UEV domain of Tsg101, is not an easily accessible description for a general audience. Consider revision of this sentence to give the audience a better idea of the nature of the UEV domain

The phrase ‘vestigial ubiquitin E2 variant (UEV) domain’ was simplified to “ubiquitin binding (UEV) domain” (line 68).

2) An ‘et al’ is missing from reference 23

Reference 23 was corrected and is now reference 32.

3) There is an underscore at the end of the last sentence/paragraph in section 2.

The underscore at the end of section 2 was removed.

4) Third sentence, first paragraph, section 3. The ‘N’ should be lowercase.

The ‘N’ in this section is now lowercase.

5) Is the term ‘Infectious’ in the title necessary?

The authors agree with Reviewer 2 that the term ‘Infectious’ is not necessary because ‘Retrovirus’ in the title implies infectivity. The title was changed to ‘Budding of a Retrovirus: Some Assemblies Required’ to reflect this.

6) I am not sure if there is a page limit for this review, but there could be a bit more ‘what we don’t know’ in the conclusion, this is just a suggestion and not required for revision

Section 5 was modified to include a discussion about what is known in the literature to contrast it with what still remains elusive (line 253 to 258).

Reviewer 3 Report

The manuscript summarized recent exciting findings of molecular interactions involved virus budding, and essential step of retroviral replication. The authors reviewed the crystal and CryoEM structures of ESCRT-1 and ALIX helical assemblies, and modeled the complex bound to Gag to explain the temporal and spatial control of the virus:host interactions during virus assembly and budding. The structure and function of short peptide motifs in host proteins, and the L domains in viral Gag protein, were discussed in details. The manuscript is very well written, providing a clear review of molecular details of interactions among ESCRT, ALIX and Gag in facilitating dissemination at sites of virus assembly.

I only have several minor concerns:

  1. Figure 1 nicely presents the ESCRT-1 and ALIX helical assemblies. The critical role of VPS28 in mediating the assembly is emphasized in the main text. A zoom-in view illustrating how VPS28 makes contacts with VPS37 and TSG101 would be helpful for the readers to visualize the structural building blocks of the large protein assembly and understand the importance of VPS28.
  2. The motif 54EKAYIKD60 in VPS28 plays an essential role in mediating ESCRT-1 helical assembly. The authors mentioned unpublished results of alanine mutational scanning, as well as the recent crystal structure determined by Flower and colleagues. It would be nice to include some molecular details on how the motif is involved in the helical interface. A figure illustrating such interactions would be helpful, too.
  3. In Figure 1E, what are the motifs/domains colored in green and yellow?
  4. First paragraph on page 4, “This VPS28 mutant …”, what does it refer to, the alanine scanning mutants or the triple mutant?
  5. Page 4, the last paragraph describes experimental results of 317FIY319 mutants. Please add reference.
  6. Page 5, lines 3-4, “These studies show for the first time that the 317FIY319 is linked to the helical assembly of ALIX …(Figure 1E). Where is the 317FIY319 in Figure 1E?
  7. Page 5, second paragraph in section 4, please add reference(s) to the processing and release defect results.
  8. Please update reference #23.
  9. Reference #24 and #29 are identical.

Author Response

Re: Reviewer #3:

Remarks to the Author

We value the authors opinion of the text and appreciate his suggestions for revision.

The manuscript summarized recent exciting findings of molecular interactions involved virus budding, and essential step of retroviral replication. The authors reviewed the crystal and CryoEM structures of ESCRT-1 and ALIX helical assemblies, and modeled the complex bound to Gag to explain the temporal and spatial control of the virus:host interactions during virus assembly and budding. The structure and function of short peptide motifs in host proteins, and the L domains in viral Gag protein, were discussed in details. The manuscript is very well written, providing a clear review of molecular details of interactions among ESCRT, ALIX and Gag in facilitating dissemination at sites of virus assembly.

I only have several minor concerns:

1. Figure 1 nicely presents the ESCRT-1 and ALIX helical assemblies. The critical role of VPS28 in mediating the assembly is emphasized in the main text. A zoom-in view illustrating how VPS28 makes contacts with VPS37 and TSG101 would be helpful for the readers to visualize the structural building blocks of the large protein assembly and understand the importance of VPS28.

This first minor concern by Reviewer 3 was an excellent suggestion and has been incorporated as an inset of Figure 1A that clearly shows the helical interface between VPS28 subunits.

2. The motif 54EKAYIKD60 in VPS28 plays an essential role in mediating ESCRT-1 helical assembly. The authors mentioned unpublished results of alanine mutational scanning, as well as the recent crystal structure determined by Flower and colleagues. It would be nice to include some molecular details on how the motif is involved in the helical interface. A figure illustrating such interactions would be helpful, too.

The inset described above that was incorporated into Figure 1A reveals the amino acid paring between residues within the 54EKAYIKD60 motif and how they interact with an adjacent VPS28 molecule as part of the helical assembly.

3. In Figure 1E, what are the motifs/domains colored in green and yellow?

The components of Figure 1E are listed in the figure legend, (Green is HIV NC, line 128, and yellow is CHMP4B, line 125). Labels were also added to Figure 1 that assist the reader.

4. First paragraph on page 4, “This VPS28 mutant …”, what does it refer to, the alanine scanning mutants or the triple mutant?

This was a helpful suggestion raised by Reviewer 3. Each mutant of VPS28 was given a name for clarity when comparing and contrasting them. The mutant by Flower et al. 2020 was named VPS28 TM (starting on line 152) while the one generated by O’Connor and colleagues was termed VPS28 EKYK (starting on line 143).

5. Page 4, the last paragraph describes experimental results of 317FIY319 mutants. Please add reference.

6. Page 5, lines 3-4, “These studies show for the first time that the 317FIY319 is linked to the helical assembly of ALIX …(Figure 1E). Where is the 317FIY319 in Figure 1E?

7. Page 5, second paragraph in section 4, please add reference(s) to the processing and release defect results.

The Reviewer requests citations for work discussed at the Nucleocapsid meeting that is not yet published and so references cannot be incorporated into the text at this time. However, the processing defects described by O’Connor and colleagues is in excellent agreement with the findings of:

Usami, Y., S. Popov, and H.G. Göttlinger, Potent rescue of human

immunodeficiency virus type 1 late domain mutants by ALIX/AIP1 depends on its

CHMP4 binding site. Journal of virology, 2007. 81(12): p. 6614-6622.

This reference (11) is described in the text (line 36).

8. Please update reference #23.

9. Reference #24 and #29 are identical.

The references mentioned by Reviewer 3 were updated in the text. Reference 23 is now 32 and reference 24 was removed.